# Brewing with Unmalted and Malted Sorghum: Influence on Beer Quality

Marius Eduard Ciocan [1], Rozália Veronika Salamon [2], Ágota Ambrus [3], Georgiana Gabriela Codină [1], Ancuța Chetrariu [1] and Adriana Dabija [1,*]

[1] Faculty of Food Engineering, Stefan cel Mare University of Suceava, 720229 Suceava, Romania; marius_ec@yahoo.com (M.E.C.); codina@fia.usv.ro (G.G.C.)

[2] Department of Food Science, Faculty of Economics, Socio-Human Sciences and Engineering, Sapientia Hungarian University of Transylvania, 530104 Miercurea Ciuc, Romania; salamonrozalia@uni.sapientia.ro

[3] Doctoral School of Chemistry, Faculty of Natural Sciences, University of Pécs, 7624 Pécs, Hungary

* Correspondence: adriana.dabija@fia.usv.ro; Tel.: +40-748-845-567

**Abstract:** One of the earliest biotechnological processes is brewing, which uses conventional raw materials like barley malt and, to a lesser extent, wheat malt. Today, adjuncts are used in the brewing of 85–90% of the world's beer, with significant regional differences. The results of this study's brewing were compared to those of beer made only from malted barley. Malted and unmalted sorghum were suggested for use in this study's brewing. In order to improve the technical mashing operation and raise output yield, commercial enzymes were introduced. The following physicochemical analyses of the finished beer were carried out in accordance with regulatory requirements: original extract (% $m/m$), apparent extract (% $m/m$), alcohol content (% $v/v$, % $m/m$), density (g/cm$^3$), turbidity (EBC), pH, color (EBC), bitterness value (IBU), oxygen content (mg/L), carbon dioxide content (g/L). A nine-point hedonic scale was used to conduct the sensory evaluation of the beer samples. Sorghum was easily included into the technological process to create a finished product that, in many ways, resembled traditional beer, making sorghum appropriate for typical beer drinkers. The laboratory brewing formula that produced the highest-quality results of all the tested variants included 60% sorghum malt and 40% unmalted sorghum: original extract 11.26% $m/m$, apparent extract 4.59% $m/m$, alcohol content 4.12% $v/v$, turbidity 0.74 EBC, $CO_2$ content 5.10 g/L. The resulting sorghum beer typically has low alcohol content, a complex, aromatic, slightly sour flavor, a mild bitter or astringent sensation, and less stable foam.

**Keywords:** gluten-free beer; unmalted sorghum; sorghum malt; sensory and nutritional properties

## 1. Introduction

Beer is one of the oldest fermented beverages and one of the most consumed low-alcohol drinks in the world, with an increase in consumption year over year [1–5]. The conventional raw materials for obtaining beer are barley malt or wheat malt, but people with known gluten sensitivity or diagnosed with celiac disease cannot consume this drink [6,7]. For a gluten-free beer, the safest method is to use gluten-free grains as raw material, such as corn, sorghum, rice, millet, etc., or pseudocereals, such as buckwheat, amaranth, and quinoa [1,2,8]. Other advantages conferred by the use of gluten-free grains in the manufacture of beer are the availability of raw materials on the local market, which also leads to the reduction of production costs, the improvement of the content in bioactive compounds and, last but not least, imparting new sensorial characteristics to the finished product which are agreeable to consumers [3,8]. More than 750 million people live in semi-arid tropical regions of Africa (Nigeria, Sudan, Burkina Faso, Ethiopia), Asia (India, China), and some parts of Central and South America, where sorghum is the most important cereal crop after corn, rice, wheat, and barley [9–12]. It is grown all over the world, the top

producing countries being India, China, Brazil, USA, and African countries, and it is the most used grain for gluten-free beer production [4,13]. Sorghum belongs, like barley and corn, to the *Poaceae* family. It is closely related to corn both in terms of genomic organization, plant shape, developmental physiology, and uses [14,15]. It is more drought-tolerant than other cereal crops, being also called the camel plant, and is therefore an important staple food in many semi-arid regions of the developing world, while in Western countries it is mainly used as animal feed [9]. Gluten-free beer is an alternative not only for gluten-intolerant people, but also for those who are interested in various new products launched on the market [1,4]. According to a report published by Fior Markets, the global gluten-free beer market will grow to USD 18.7 billion by the year 2025, with an annual growth rate of 16.3% [4,16].

There is much information in the literature about the use of unmalted sorghum or sorghum malt in brewing. Research to date has focused on optimizing the sorghum brewing process and obtaining a finished product with sensory characteristics that meet consumer acceptability [2]. The studies carried out focused on the possibility of using sorghum as an adjuvant to obtain beer, either in ground or extruded form in a proportion of 40–60% compared to barley malt [2,3,17,18] or used in beer manufacturing at 100% sorghum malt [13,18–20]. The presence of sorghum beers on the market is recognition of the possibility of using this grain in brewing. It is essential to obtain high-quality sorghum malt or it is advised to combine it with other grains to enhance the completed product's sensory qualities [21,22]. Sorghum, both as malted sorghum and as an adjuvant, has historically been used in Africa as the primary raw material in brewing [23–25]. Currently, sorghum beer in these countries is obtained by traditional and industrial methods, by spontaneous fermentation or directed fermentation, known under different local names depending on the region or ethnic group [26–29]. In particular, sorghum beer is also known as kefir, pito, or burukutu in Ghana, Togo, and Nigeria, and bantu or utshwala in South Africa. Burkina Faso, Mali, Senegal, and Côte d'Ivoire call it dolo, doro, or tchapalo, whereas in Benin, Togo, and northern Nigeria it is tchoukoutou or chakpalo, and in Nigeria and Ghana, otika. It is bili-bili, ambga, red kapsiki, or dora-bonga in the Central African nations of Chad and Cameroon, Doro, chibuku, uthwala or chikokivana in Zimbabwe; omalovu, tombo, or epwaka in Namibia; ikigage or awarwa in Rwanda; merissa in Sudan; talla in Ethiopia; mtama in Tanzania; munkoyo in Congo and Zambia; and busaa in Kenya [28,30,31]. In these countries, beer has sociocultural and nutritional value, so it plays a central role in people's cultures and represents an important part of the diet for an increasing part of the urban population [1,28,29]. In Africa, sorghum beer is perceived by consumers as having therapeutic qualities due to its bioactive compounds, antioxidant activity, and high content of phenolic compounds [29,31–34]. Unlike industrially produced sorghum beer, traditional beer is not filtered and stabilized, it is consumed in a state of active fermentation, it has a very short shelf life of 24–72 h, it is opaque, with a slightly sour taste, rich in bioactive compounds and a relatively low alcohol concentration [3,28]. The production of clear lagers from sorghum has also been reported in other parts of the world. In Mexico, a lager-type beer was obtained from sorghum. Sorghum has been used as an adjuvant in the brewing of lager beer in the USA since the 1980s. It might seem more practical to use unmalted sorghum and commercial enzymes due to the issues with malted sorghum, including the development of insufficient diastatic power, high gelatinization temperatures, high mash viscosity, low content of free amino acids, limited protein modification due to low proteolytic activity, high malting costs, high malting losses, as well as the need to supplement the unmalted sorghum with exogenous enzymes [1,35–37]. Espinosa-Ramírez et al. successfully produced lagers from different types of sorghum malts using β-amylase or amyloglucosidase [38]. Sorghum beer finished product generally has a low alcohol concentration, with a complex, aromatic, slightly sour taste, a slight bitter or astringent sensation, shows less foam stability, and a short shelf life [39–41].

The objective of this study was to assess the effect of using malted and unmalted sorghum in the production of beer by comparing the results to beer produced exclusively from barley malt.

## 2. Materials and Methods

### 2.1. Raw Materials

Sorghum (*Sorghum bicolor*) from the 2021 crop was grown in the eastern part of Romania for the tests, together with sorghum malt from the USA (Pennsylvania Craft Malt). Romanian barley from the 2020 crop and pilsner-style barley malt were used in the comparison analysis. Amarillo Yachima Chief hops from the Yakima, Washington, United States, 2019 harvest, Fermentis brewer's yeast type 74/30 from Marquette-lez-Lille, France, and an enzyme preparation from the Novozyme Company from Bagsvaerd, Denmark, were also utilized to brew the beer in a lab setting. The ingredients for the manufacturing recipes were first weighed, put together, and then ground.

### 2.2. Obtaining Beer in Laboratory Conditions

According to EBC method 4.5.1, all the mash samples were obtained in Mash Bath R12 with PC connection (1-CUBE, Havlckuv Brod, Czech Republic) (European Brewery Convention, Analytica EBC, 2004) [42]. The quality of the malt was determined using physical and chemical assessments of wort made using the congress method (for samples of barley malt) and the modified congress method (for samples of sorghum malt). Table 1 shows the variations of the tested brewing recipes, with variant CS denoting brewing with 100% barley malt. In these tests, varying amounts of sorghum malt and unmalted sorghum were utilized. Based on suggestions from expert literature and practical experience, an enzyme preparation with thermostable α-amylase amounts from Table 1 was added (the ideal quantity of the enzyme employed industrially for manufacturing beer from unmalted and malted barley and barley is 4%). At a temperature of 35 °C, the enzyme was added at the beginning of the mashing process.

**Table 1.** Research with different brewing mashing recipes.

| Ingredient | Brewing Recipe | | | | | |
| --- | --- | --- | --- | --- | --- | --- |
| | CS | S1 | S2 | S3 | S4 | S5 |
| Barley malt | 100 | - | - | - | - | - |
| Sorghum malt, % | - | 100 | 70 | 60 | 50 | - |
| Unmalted sorghum, % | - | - | 30 | 40 | 50 | 100 |
| Termamyl classic enzyme preparation, % | 0 | 4 | 4 | 4 | 4 | 4 |

The beers made with sorghum (S1, S2, S3, S4, S5) were compared with those made exclusively from barley malt (CS). All laboratory tests, including those on different mash methods and brewing, had at least three replications.

#### 2.2.1. Mashing

The unmalted and malted sorghum samples were ground for this purpose in a laboratory disc mill of the Perten LM 3310 (Cheltenham, UK) type with a 0.20 mm gap between the grinding discs. The barley malt had been subjected to the Congress method (EBC procedure 4.5.1) (Figure 1). Sorghum malt and sorghum as an unmalted component were prepared using the modified congress method, as indicated in Figure 2.

With a negative iodine test, which shows that iodine produces a yellow color rather than a violet one, the saccharification rate was determined. The sample containing unmalted sorghum did not saccharify after 60 min of retaining the variant 0 at 72 °C. The results of this experiment recommend trying using enzyme preparations to accelerate the saccharification of mash. It was necessary to conduct more research to determine the optimal enzyme dosage for brewing.

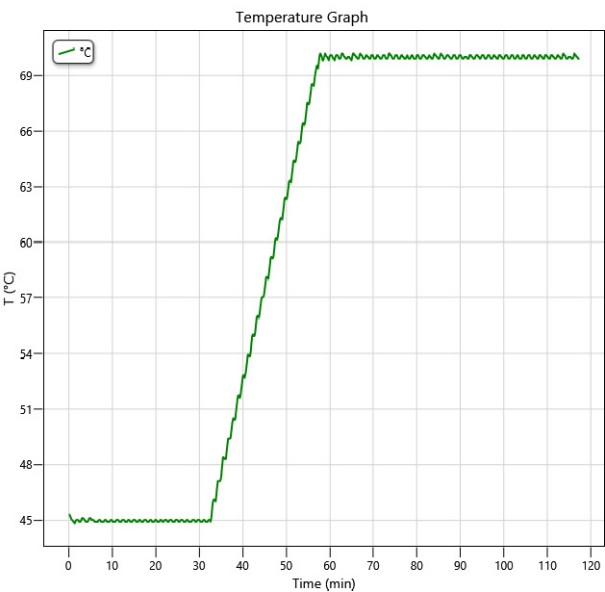

**Figure 1.** Congress mashing method.

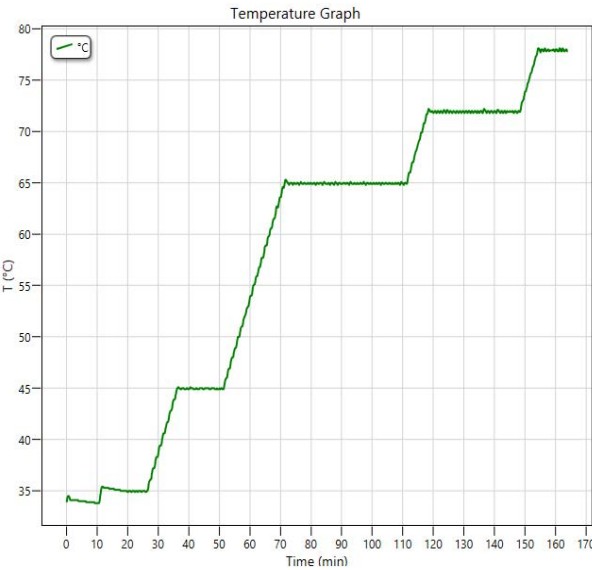

**Figure 2.** Modified congress mashing method.

### 2.2.2. Mash Filtration

Following mashing, distilled water was used to dilute the contents of each flask mash beaker to 450 g, which was then homogenized and put through a pleated folding filter with diameter of 290 mm separately. To ensure that the resultant congress wort had a high degree of clarity, the first 100 mL of wort was handed over to the filter. The wort's quality characteristics were assessed separately before boiling.

### 2.2.3. Wort Boiling

Congress wort (2–2.2 L) from three mash beakers was boiled for an hour in Erlenmeyer flasks with hops (1.52–1.54 g Amarillo hops, 7.8% alpha bitter acids). Twenty IBU was the target wort bitterness. After boiling, the wort (1.5–1.7 L) was cooled to 20 °C, allowed to sit for 30 min to allow the hot trub to settle, and then decanted into 2 L glass fermentation vessels with $CO_2$ exhaust valves. The wort was examined prior to yeast inoculation.

### 2.2.4. Fermentation

The wort (1.5 L) was cooled to 12 °C before being inoculated with 25 g of yeast type 74/30 Fermentis, the quality of which had been assessed using the Nucleocounter YC-100 (ChemoMetec A/S, Lillerød, Denmark), with a total cell count of $24 \times 10^8$/mL biomass, including 1.42% dead cells. After the primary fermentation took place in glass fermentation vessels of type Erlenmayer with 2 L at a temperature of to 12 °C for 6 days, with the possibility of digital regulation, the young beer was bottled and subjected to secondary fermentation and maturation for 30 days at 4 °C in the same industrial refrigerator. We repeated each laboratory brewing experiment at least three times.

### 2.3. Methods of Analysis

#### 2.3.1. Analysis of Raw Materials and Wort

According to the newest editions of Analytica EBC by the European Brewery Convention, the raw materials and wort qualities of the barley and sorghum samples were determined. The European Brewery Convention's latest versions of Analytica EBC were used to determine the moisture (EBC 4.2) [43], foreign matters (EBC 4.22) [44], thousand corn weight, protein content, and starch content for the samples of barley and sorghum. The following results were recorded from the wort: saccharification time rate (EBC 4.5.1), speed of filtration time (EBC 4.5.1), color (EBC 4.7.1), pH of wort, original extract (EBC 8.3), extract yield (%/dry weight.; EBC 4.2), protein content (%/dry weight.; EBC 4.3.1), soluble nitrogen (mg/L; EBC 4.9.1), soluble nitrogen (relative to dry weight from malt), FAN (mg /100 g dry weight; EBC 4.10) with spectrophotometer Hach Lange DM 6000 (Hach Lange GmbH, Düsseldorf, Germany), apparent degree of fermentation (%, EBC 4.11.2) with Anton Paar Alex (Anton Paar Austria GmbH, Graz, Austria) [26], saccharification rate and filtration speed (EBC 4.5.1) [43], color of wort (EBC 4.7.1) [45] and free amino nitrogen in wort (FAN, EBC 8.10.1) [46] with spectrophotometer Hach Lange DM 6000 (Hach Lange GmbH, Düsseldorf, Germany); pH of wort (EBC 8.17) WTW Inolab 720 table pH meter (Xylem Analytics, Ingolstadt, Germany) [47], original extract (EBC 8.3) [48], extract of malt, congress mash (EBC 8.3) [42], fermentability, final attenuation of laboratory wort from malt (EBC 4.11.2) [49] with Anton Paar Alex (Anton Paar Austria 158 GmbH, Graz, Austria), total nitrogen of malt (EBC 4.3.1) [50], soluble nitrogen of malt (EBC 4.9.1) [51] with Velp Scientifica UDK129 (Velp Scientifica Srl, Usmate, Italy).

Analytical conclusions were drawn from each mashing beaker, and the results are shown along with the mean and standard deviation. Every sample's results were analyzed in triplicate, and for the purpose of this study report, the mean values were taken into account.

#### 2.3.2. Analysis of Beer

According to the latest versions of Analytica EBC's standard operating procedures, the following physicochemical analyses for the finished beer were carried out: the original extract and density (EBC 9.43.2) [52], the alcohol content (% $v/v$, % $m/m$; EBC 9.2.1) [53], turbidity, EBC 9.29) [54], pH (EBC 9.35) [55], $CO_2$ content (EBC 9.28.3) [56], $O_2$ content (EBC 9.37.1) [57], calories kJ/100 mL (EBC 9.45) [58], Anton Paar modular DMA meter (Anton Paar Austria 158 GmbH, Graz, Austria), color (EBC 9.6) [59], and bitterness (EBC 9.8) [60]. The mean value was used in the present study after three determinations from the same sample were analyzed simultaneously.

#### 2.3.3. Sensory Analysis

Sensory evaluation was performed in the Sensory Analysis Laboratory of the Ștefan cel Mare University, Faculty of Food Engineering (Suceava, Romania) using for this purpose 55 semi-trained evaluators who were frequent beer drinkers (34 women and 21 men, 20–27 years old). Subjects were recruited from the Faculty of Food Engineering, Ștefan cel Mare University with knowledge in sensory science and experience with formal sensory evaluation. Evaluators were considered semi-trained panelists due to the fact that they

were chosen based on their sensory acuity for which they were screened. More, all the participants were knowledgeable on beer characteristics and technology and also sensory analysis due to the fact that they took courses and worked in laboratories during their undergraduate programs. Beer samples were presented at 8 °C in an amount of 70 mL in glass cups, identified with three-digit numbers. Between samples, the evaluators used water and unsalted crackers to cleanse their mouths. For the beers' evaluation, a nine-point structured hedonic scale ranging from "dislike extremely" = 1 to "like extremely" = 9 was used. The sensory characteristics of the beer were examined in the fallowing order: general acceptability, appearance, color, general taste, aroma, bitterness, body, mouthfeel, and carbonation. The tasting space and equipment complied with EBC 13.2 [61].

### 2.3.4. Statistical Analysis

All data analyses were performed in triplicate and results were expressed as mean ± standard deviation. Physicochemical and sensory characteristics of experimental samples were assessed by one-way ANOVA and Tukey's HSD at alpha = 0.05 was used to test for significant differences between mean data. Using the Design of Experiments software (DOE) (Design Expert, trial edition, Stat-Ease, Inc., Minneapolis, MN, USA), the data were plotted illustrating the fluctuation of beer parameters according to the varied levels of sorghum flour and sorghum malted flour. The mean value of the sensory data was graphically displayed by the assessors using Microsoft Excel. Principal component analysis (PCA) was performed in order to analyze the differences and correlations between beer samples and their physicochemical characteristics. For this purpose, XLSTAT 2021.2.1 software was used.

## 3. Results and Discussion

### 3.1. Evaluation of Sorghum and Sorghum Malt Quality

The potential of unmalted and malted sorghum as raw material in brewing was initially ascertained through this research. The qualitative characteristics of sorghum and barley as unmalted raw materials are compared in Table 2.

**Table 2.** The main physicochemical characteristics of barley and sorghum used in these experiments.

| Characteristic | Values Obtained for: | |
| --- | --- | --- |
|  | **Barley** | **Sorghum** |
| Moisture, % | 11.00 ± 0.20 | 10.20 ± 0.15 |
| Thousand corn weight, g | 39.87 ± 0.50 | 28.40 ± 0.20 |
| Foreign matters, total, % | 3.23 ± 0.10 | 4.05 ± 0.10 |
| Protein content, %/d.w. | 10.80 ± 0.30 | 10.30 ± 0.25 |
| Starch content, %/d.w. | 62.00 ± 0.40 | 72.40 ± 0.30 |

Results represent mean values ± standard deviation (SD), *n* = 3.

Sorghum grains' moisture content (10.20%) was found to be lower than barley grains' (11.00%) moisture content. This small discrepancy could be the result of the way it was transported or stored. Due to the size of the grains, sorghum has a substantially lower mass per 1000 grains than barley; 28.40% to be exact. This indicator's size is related to the volume of the extract. The losses during conditioning of sorghum, the raw material, will reflect the fact that sorghum contains more foreign matter than barley. Impurities may facilitate the growth of microorganisms, resulting in malt of inferior quality. The evidence from the specialized literature agrees with the other indicators. As a result, the 10.8% protein content is comparable to that reported by Schnitzenbaumer and Arendt (9–13.5%) [9], Shegro et al. (8.08–15.26%) [62], Rashwan et al. (10.3%) [63], and Ogu et al. (9.6–10.9%) [64]. The figure of 72.40% for the starch content of the sorghum utilized in our research is consistent with other studies' findings of 61.0–74.8% [9], 72.09% [42], and 72.41% [65]. The quality characteristics of the sorghum malt and barley malt utilized in the study are listed in Table 3.

**Table 3.** Results of the physicochemical characteristics analysis of barley malt and sorghum malt.

| Characteristic | Barley Malt | Sorghum Malt |
|---|---|---|
| Moisture, % | 5.10 ± 0.10 | 8.64 ± 0.15 |
| Color, EBC-unit | 3.60 ± 0.10 | 4.80 ± 0.10 |
| Extract of congress wort, °P | 8.90 ± 0.30 | 8.60 ± 0.40 |
| Saccharification rate, min. | 9.00 ± 0.40 | >30 |
| Filtration speed, min. | 14.00 ± 1.00 | >50 |
| Wort pH | 6.05 ± 0.10 | 6.17 ± 0.10 |
| Wort appearance | clear | weak opal |

Results represent mean values ± standard deviation (SD), *n* = 3.

The sorghum malt demonstrated appropriate quality indices for beer manufacturing, as can be seen from the comparative analysis of the data that have been presented. Sorghum malt showed similar pH and wort extract concentration to barley malt, while recording greater values for malt moisture and wort color. It was suggested to use enzyme preparations to shorten these times in order to create beer under laboratory settings because the saccharification and filtration times were longer than they were for barley malt. The data obtained for sorghum malt are close to those obtained by Rubio-Flores et al. for the color of the beer wort (4.62 unit, EBC) [35], for the duration of saccharification to those determined by Coulibaly et al. (57.39 min) [30] and by Gumienna and Górna (60 min) [4], for moisture to those obtained by Yafetto et al. (8.34%) [66], and for the content in the extract and the pH of the wort to those determined by Tokpohozin et al. [67].

*3.2. Experimental Brewing with Unmalted and Malted Sorghum under Laboratory Conditions*

The mix of ingredients was combined with water to create the wort before the brewing process began. Gelatinization, liquefaction, and saccharification were the stages of the fermentation process. The full transformation of the starch in the plasma was confirmed by the iodine test. A good malt saccharifies in less than 10 min; a longer duration is the result of insufficient starch breakdown. It is well known that the gelatinization temperature of a starch is a crucial element in starch degradation. [68].

The use of sorghum malt in brewing was known to cause certain issues because of the low amylolytic activity, which was insufficient for a complete saccharification, the high gelatinization temperature, as well as the low quantity of free amino nitrogen. Sorghum has a lower β-amylase activity than barley malt, but a higher α-amylase activity. Reduced enzyme activity will result in a decrease in the amount of fermentable carbohydrates produced, an increase in the concentration of dextrins, and ultimately an increase in viscosity [38,69]. Kafirins restrict gelatinization temperature [70]. As a result, only a portion of starch is hydrolyzed into fermentable sugars. Therefore, using sorghum in brewing necessitates a suitable malting procedure to prevent technological issues. Otherwise, using exogenous enzymes to make sorghum beers is advised [70,71].

When milling sorghum, Espinosa-Ramrez et al. (2013) investigated the impact of adding amyloglucosidase or β-amylase to increase the alcohol concentration [38]. Amyloglucosidase was utilized by Urias-Lugo and Salvidar (2005), which improved wort yield and filtration rate and, as a result, produced a higher percentage of ethanol. Sorghum beer had a 1.1% lower alcohol concentration than barley malt beer, nonetheless [72]. Amyloglucosidase was added; however, it had no effect on the color, pH, or FAN content [72]. *Aspergillus oryzae*, which has been demonstrated to enhance sorghum's malting capabilities and increase wort and beer yields when compared to ordinary sorghum malt, can also be added to mitigate these inadequacies. While there were no differences for β-amylase, α-amylase was positively influenced by the use of this adjuvant [35,70].

The main problems in brewing with sorghum are the lower diastatic strength of its malt, especially the deficiency in β-amylase activity and the higher gelatinization temperature of sorghum starch compared to barley starch [35]. In research conducted by Espinosa-

Ramirez et al. (2013), lagers were successfully produced from different types of sorghum malts and gluten-free adjuvants supplemented with β-amylase or amyloglucosidase [38].

Table 4 lists the results of the analyses performed on the beer wort produced by all five of the brewing recipes that were investigated. These findings were compared with a beer wort sample produced from barley malt.

**Table 4.** Physicochemical properties of wort.

| Characteristic | Brewing Recipe Variants | | | | | |
|---|---|---|---|---|---|---|
| | CS | S1 | S2 | S3 | S4 | S5 |
| Filtration speed, minutes | 20 ± 2.00 [a] | 40.00 ± 1.00 [b,A] | 40.00 ± 1.00 [b,A] | 40.00 ± 1.00 [b,A] | 40.00 ± 1.00 [b,A] | 40.00 ± 1.00 [b,A] |
| Saccharification rate, minutes | 10 ± 1.00 [a] | 10.00 ± 2.00 [a,A] | 10.00 ± 2.00 [a,A] | 10.00 ± 2.00 [a,A] | 10.00 ± 2.00 [a,A] | 10.00 ± 2.00 [a,A] |
| Extract of congress wort, °P | 8.95 ± 0.30 [a] | 8.60 ± 0.50 [a,A] | 7.80 ± 0.50 [b,A] | 7.40 ± 0.50 [b,B] | 7.30 ± 0.50 [b,B] | 6.60 ± 0.50 [c,C] |
| Extract yield, % d.w. | 84.05 ± 0.20 [a] | 83.24 ± 0.40 [a,A] | 74.35 ± 0.30 [b,B] | 69.98 ± 0.25 [b,C] | 68.93 ± 0.34 [c,C] | 61.47 ± 0.28 [c,C] |
| Color, EBC | 3.40 ± 0.10 [a] | 3.80 ± 0.20 [b,A] | 3.90 ± 0.20 [b,A] | 3.80 ± 0.10 [b,A] | 3.90 ± 0.10 [b,A] | 3.50 ± 0.10 [a,B] |
| pH | 6.06 ± 0.02 [a] | 6.17 ± 0.05 [a,B] | 6.25 ± 0.05 [b,A] | 6.26 ± 0.05 [b,A] | 6.22 ± 0.05 [b,A] | 6.28 ± 0.05 [b,A] |
| Total proteins, % d.w. | 10.22 ± 0.05 [a] | 8.35 ± 0.07 [b,A] | 8.43 ± 0.08 [b,A] | 8.46 ± 0.06 [b,A] | 8.49 ± 0.05 [b,A] | 8.63 ± 0.06 [b,B] |
| Soluble nitrogen, mg/L | 678 ± 0.58 [a] | 193.90 ± 0.65 [c,A] | 206.90 ± 0.76 [c,B] | 196.60 ± 0.57 [c,A] | 187.30 ± 0.84 [c,A] | 195.70 ± 0.74 [c,A] |
| Total nitrogen, % d.w. | 1.54 ± 0.10 [a] | 1.34 ± 0.20 [b,A] | 1.35 ± 0.10 [b,A] | 1.35 ± 0.20 [b,A] | 1.36 ± 0.10 [b,A] | 1.38 ± 0.10 [b,A] |
| FAN, mg/100 g | 116.28 ± 0.66 [a] | 51.67 ± 0.63 [b,B] | 48.23 ± 0.60 [c,A] | 60.81 ± 0.55 [b,C] | 49.53 ± 0.68 [c,A] | 43.83 ± 0.58 [c,B] |
| Kolbach Index | 39.04 ± 0.50 [a] | 21.96 ± 0.64 [c,A] | 22.77 ± 0.50 [c,A] | 21.39 ± 0.44 [c,A] | 20.27 ± 0.48 [c,A] | 20.50 ± 0.62 [c,A] |
| Apparent degree of fermentation,% | 84.96 ± 0.40 [a] | 58.13 ± 0.50 [c,B] | 60.25 ± 0.50 [b,A] | 60.81 ± 0.40 [b,A] | 61.11 ± 0.40 [b,A] | 63.63 ± 0.50 [b,B] |

Results represent mean values ± standard deviation (SD), *n* = 3; different letters indicate (a, b, c) that the result presents significant differences from the control (CS) at *p* < 0.01 level; different letters indicate (A, B, C) that the result presents significant differences between the variants (S1, S2, S3, S4, S5) at *p* < 0.01 level. Variant: CS—100% barley malt; S1—100% sorghum malt; S2—70% sorghum malt and 30% unmalted sorghum; S3—60% sorghum malt and 40% unmalted sorghum; S4—50% sorghum malt and 50% unmalted sorghum; S5—100% unmalted sorghum.

The obtained beer wort followed the hops-infused boiling process. Table 5 provides an overview of the physicochemical properties of sorghum wort after it has been boiled with hops.

**Table 5.** Physicochemical characteristics of boiled wort.

| Characteristic | Brewing Recipe Variant | | | | | |
|---|---|---|---|---|---|---|
| | CS | S1 | S2 | S3 | S4 | S5 |
| Wort extract, °P | 11.50 ± 0.10 [a] | 11.30 ± 0.20 [a,A] | 11.10 ± 0.10 [a,A] | 10.50 ± 0.10 [b,B] | 9.50 ± 0.20 [b,B] | 8.50 ± 0.10 [c,C] |
| Color, EBC | 4.60 ± 0.10 [a] | 8.10 ± 0.10 [d,A] | 8.40 ± 0.10 [d,A] | 8.90 ± 0.20 [d,A] | 6.92 ± 0.10 [c,C] | 5.41 ± 0.10 [b,C] |
| pH | 6.01 ± 0.05 [a] | 6.14 ± 0.05 [b,B] | 6.07 ± 0.05 [a,A] | 6.15 ± 0.05 [b,B] | 6.03 ± 0.05 [a,A] | 6.11 ± 0.05 [b,B] |
| Bitterness value, IBU | 64.70 ± 0.36 [a] | 65.2 ± 0.40 [a,A] | 61.20 ± 0.32 [c,C] | 64.40 ± 0.52 [a,A] | 64.90 ± 0.48 [a,A] | 63.6 ± 0.54 [b,B] |

Results represent mean values ± standard deviation (SD), *n* = 3; different letters indicate (a, b, c, d) that the result presents significant differences from the control (CS) at *p* < 0.01 level; different letters indicate (A, B, C) that the result presents significant differences between the variants (S1, S2, S3, S4, S5) at *p* < 0.01 level.

All samples included in the study had a marginally lower pH after 60 min of boiling when compared to the identical samples without boiling. The boiling operation caused water to evaporate, increasing the extract content in all samples. All evaluated samples showed an increase in hue after boiling of between 3.4 and 3.9 EBC at 4.5–8.9 EBC.

After being cooled to 12 °C and inoculated with yeast, the young beer was bottled and underwent a 30-day secondary fermentation and maturation process. The initial fermentation took place for six days. Table 6 lists the physical and chemical characteristics of the finished beer.

Comparing the beers produced from the five variants and the control sample, real extract, apparent extract, and alcohol percentage differed, although color, pH, bitterness, and carbon dioxide concentration only slightly differed. A significant identifying characteristic was the amount of ethanol present, which was affected by the use of unmalted and malted sorghum.

**Table 6.** Physicochemical characteristics of beer—finished product.

| Characteristic | Brewing Recipe Variant | | | | | |
|---|---|---|---|---|---|---|
| | CS | S1 | S2 | S3 | S4 | S5 |
| Original extract, % $m/m$ | 11.10 ± 0.05 [a] | 10.40 ± 0.05 [b,A] | 11.38 ± 0.04 [c,B] | 11.26 ± 0.05 [d,C] | 9.92 ± 0.04 [e,D] | 8.90 ± 0.04 [f,E] |
| Apparent extract, % $m/m$ | 2.40 ± 0.02 [a] | 4.14 ± 0.06 [b,A] | 4.28 ± 0.04 [c,B] | 4.59 ± 0.04 [d,C] | 3.11 ± 0.04 [e,D] | 2.68 ± 0.03 [f,E] |
| Alcohol content, % $v/v$ | 4.75 ± 0.04 [a] | 3.32 ± 0.05 [b,A] | 3.78 ± 0.05 [c,B] | 4.12 ± 0.06 [d,C] | 3.04 ± 0.04 [e,D] | 2.81 ± 0.05 [f,E] |
| Alcohol content, % $m/m$ | 3.70 ± 0.02 [a] | 2.58 ± 0.05 [b,A] | 2.94 ± 0.05 [c,B] | 3.20 ± 0.06 [d,C] | 2.38 ± 0.05 [e,D] | 2.20 ± 0.06 [f,E] |
| Density, g/cm$^3$ | 1.00835 ± 0.0002 [a] | 1.01444 ± 0.0001 [b,A] | 1.01492 ± 0.0001 [c,B] | 1.01616 ± 0.0002 [d,C] | 1.01036 ± 0.0002 [e,D] | 1.00869 ± 0.0002 [e,E] |
| Turbidity, EBC | 0.76 ± 0.01 [a] | 1.56 ± 0.02 [b,A] | 1.12 ± 0.01 [c,B] | 0.74 ± 0.02 [a,C] | 1.49 ± 0.01 [d,D] | 1.43 ± 0.03 [a,E] |
| pH | 4.60 ± 0.04 [a] | 4.69 ± 0.05 [a,A] | 4.89 ± 0.05 [b,B] | 4.75 ± 0.05 [c,B] | 4.62 ± 0.05 [a,A] | 4.70 ± 0.05 [d,A] |
| Color, EBC | 5.20 ± 0.12 [a] | 5.60 ± 0.15 [b,A] | 6.60 ± 0.12 [c,B] | 6.80 ± 0.18 [c,B] | 5.10 ± 0.11 [a,C] | 4.50 ± 0.12 [d,D] |
| Bitterness value, IBU | 25.30 ± 0.50 [a] | 24.80 ± 0.58 [a,A] | 32.20 ± 0.52 [b,B] | 25.80 ± 0.70 [a,A] | 25.40 ± 0.85 [a,A] | 26.50 ± 0.78 [c,C] |
| $CO_2$, g/L | 4.90 ± 0.04 [a] | 4.82 ± 0.05 [a,A] | 4.80 ± 0.07 [a,A] | 5.10 ± 0.05 [b,B] | 4.90 ± 0.07 [a,A] | 4.87 ± 0.08 [a,A] |
| $O_2$, mg/L | 1.20 ± 0.01 [a] | 3.15 ± 0.01 [b,A] | 2.44 ± 0.01 [c,B] | 3.22 ± 0.03 [b,A] | 3.20 ± 0.02 [b,A] | 3.23 ± 0.01 [b,B] |
| Calories, kJ/100 mL | 145 ± 0.50 [a] | 139 ± 0.55 [a,A] | 152 ± 0.53 [b,B] | 161 ± 0.72 [a,A] | 115 ± 0.84 [a,A] | 105 ± 0.76 [c,C] |

Results represent mean values ± standard deviation (SD), $n = 3$; different letters indicate (a, b, c, d, e, f) that the result presents significant differences from the control (CS) at $p < 0.01$ level; different letters indicate (A, B, C, D, E) that the result presents significant differences between the variants (S1, S2, S3, S4, S5) at $p < 0.01$ level.

Unmalted sorghum (50% of the total grain wet weight) and barley malt (50% of the total grain wet weight) were used as the primary ingredients in the production of beer by Agu et al. (2002) on a pilot scale (1000 L), which was flocculated with rests at 50 °C, 60 °C, and 65 °C [73]. According to sensory analysis, there were no appreciable variations in the odor, taste, or clarity between the commercial barley beer, the control beer, and the sorghum beer. However, in terms of taste, flavor, and foam stability, sorghum beer was discovered to be significantly different from both beers. Figure 3 illustrates the estimated total influence of sorghum malt and unmalted sorghum on the beer's original extract, apparent extract, alcohol content, density, pH, and turbidity, and Figure 4 shows the effects of unmalted and malted sorghum on the beer's bitterness value, color, $O_2$ content, and $CO_2$ content.

The sensory characteristics of beer samples are shown in Table 7. Mean general acceptability score ranged between 6.41 and 8.42. According to our data, all beer samples were liked by the panelists from "like slightly" to "like very much". No significant differences ($p < 0.05$) were obtained for general acceptability between beers S3 and S1. It seems that these types of beers were also similar in their aroma and mouthfeel sensory characteristics. These types of beers with sorghum malt in their recipe were the most appreciated ones after the control sample beer. These data are in agreement with those reported by Owuama [74], who concluded that the character of beers from barley and sorghum malt is somehow comparable. According to him, these types of beers are acceptable with slight differences in flavor, taste, and color. Sample CS was the most appreciated one, receiving the highest score for sensory characteristics of appearance, aroma, general taste, carbonation, body, and mouthfeel. The beer sample S2 was less appreciated, receiving the lowest scores for sensory characteristics of color, aroma, general taste, bitterness, carbonation, body, and mouthfeel. All the beer samples were very well appreciated for carbonation, with mean values ranging between 8.31 and 8.81. This fact was explainable since it has been reported that sorghum grain contains an amylase fraction, which is involved in starch hydrolysis during mashing [75]. It presents a lower β-amylase activity than barley malt but a higher α-amylase one [3]. Additionally, the main compound of sorghum grain is starch, its content being slightly higher than that from barley [76]. This will lead to a high production of fermentable sugars and therefore carbon dioxide, especially if it is combined with barley or sorghum malt. The appearance sensory characteristics were less appreciated for beer samples with high amounts of sorghum flour incorporated in their recipe. However, these beer samples were well appreciated for their color. This fact was explainable since sorghum kernel pigmentation has more whiteness, which may not affect the beer color value. Being a blonde beer type and taking into account that sorghum addition did not alter the beer color very much, this led to a good appreciation of this sensory characteristic by the panelists. The highest sensory bitterness value was obtained for the control sample. It seems that

sorghum addition in beer recipe will lead to a significant decrease ($p < 0.05$) in this value. According to Adetunji et al. [77], the white sorghum has a sweet, which may have decreased the bitterness value of beer samples in which it was incorporated.

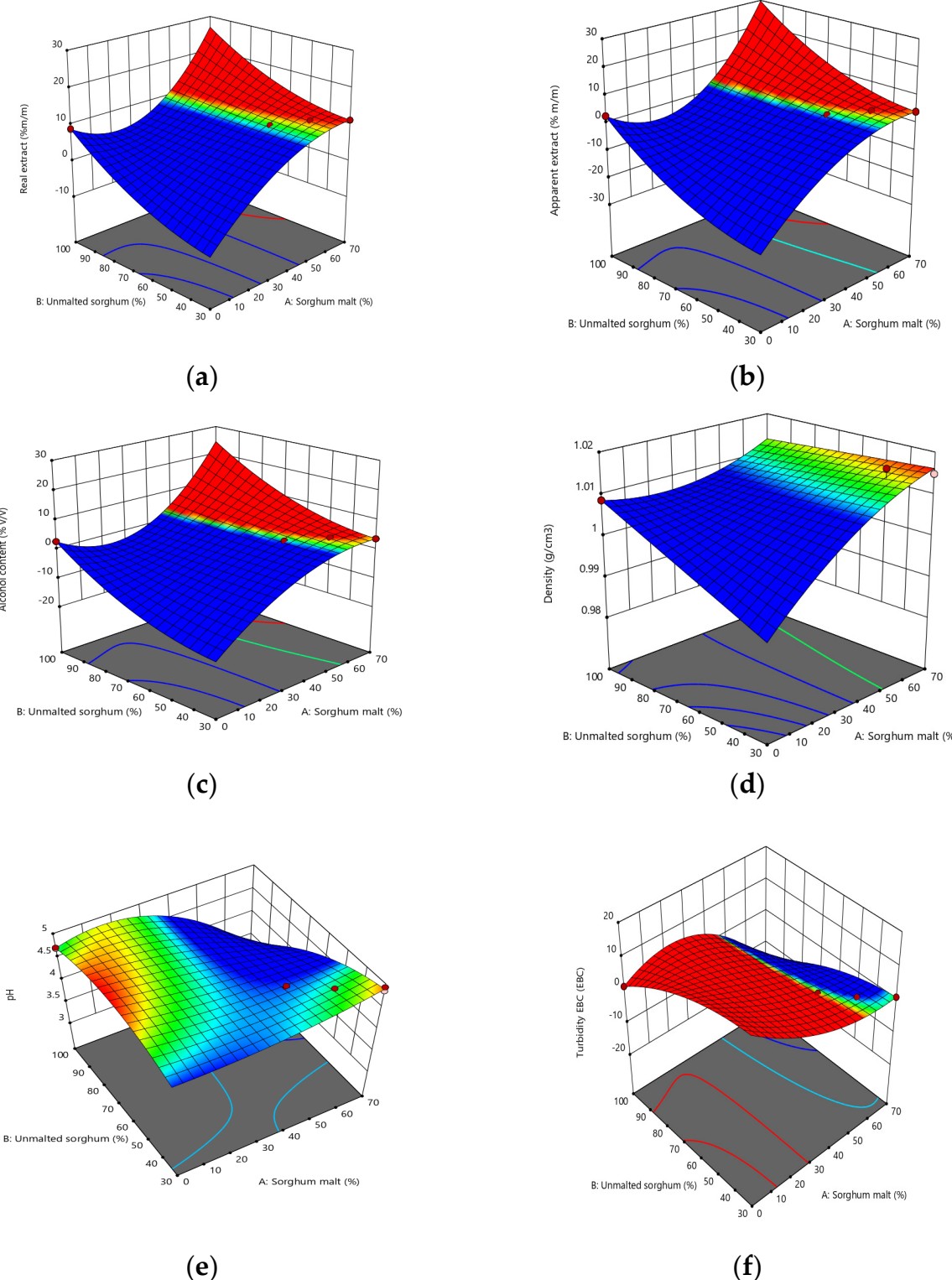

**Figure 3.** The graphical representations of the original extract (**a**), apparent extract (**b**), alcohol content (**c**), density (**d**), pH (**e**), and turbidity (**f**) as affected by the levels of unmalted sorghum and sorghum malt used in the beer recipe.

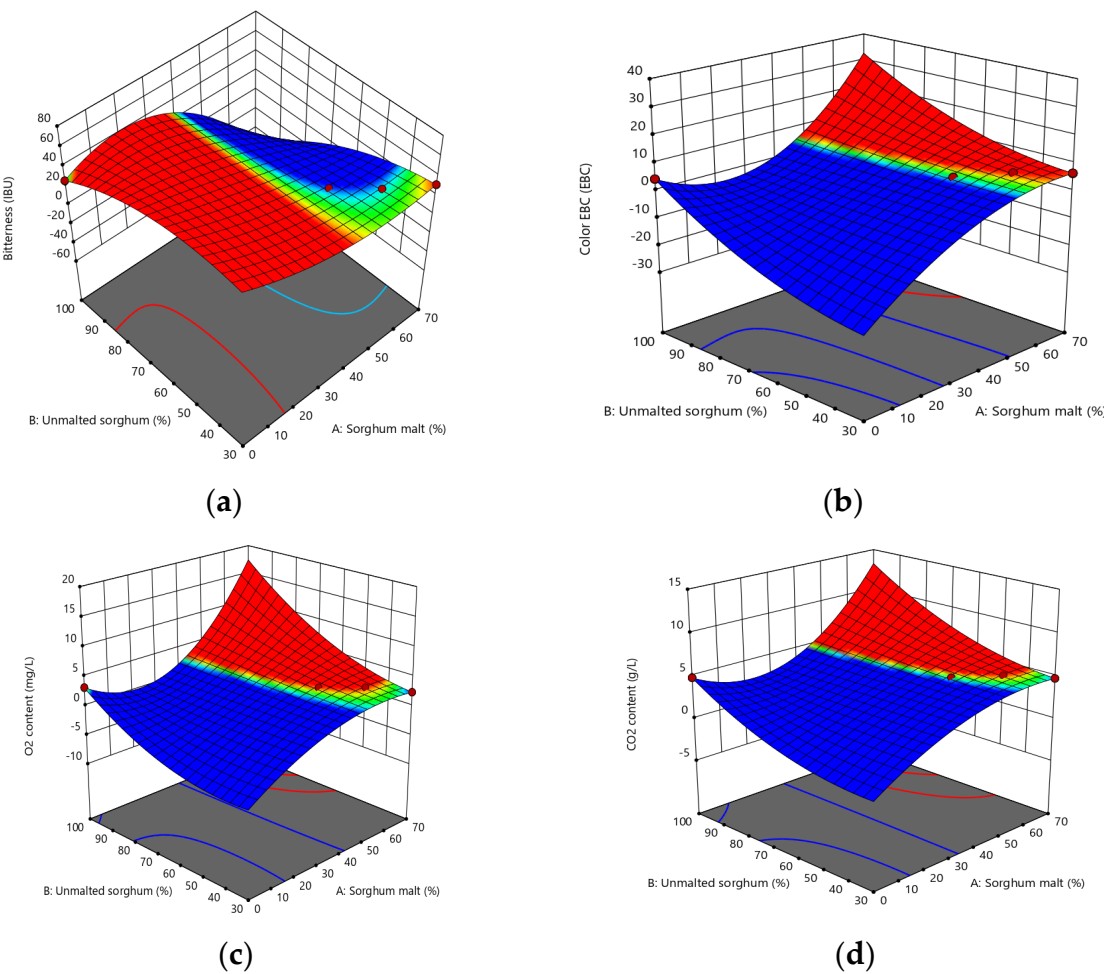

**Figure 4.** The graphical representations of the bitterness value (**a**), color (**b**), $O_2$ content (**c**), and $CO_2$ content (**d**) as affected by the levels of unmalted sorghum and sorghum malt used in the beer recipe.

**Table 7.** Sensory characteristics of the beer samples.

| Characteristic | Brewing Recipe Variant | | | | | |
|---|---|---|---|---|---|---|
| | CS | S1 | S2 | S3 | S4 | S5 |
| Appearance | 8.71 ± 0.17 [e] | 7.04 ± 1.18 [c] | 7.06 ± 0.33 [b] | 8.60 ± 0.13 [e] | 6.62 ± 1.28 [a] | 6.82 ± 1.03 [b] |
| Color | 7.52 ± 1.00 [d] | 7.12 ± 1.40 [b] | 6.71 ± 0.23 [a] | 7.36 ± 0.57 [bc] | 7.31 ± 0.95 [c] | 8.10 ± 0.23 [d] |
| Aroma | 8.61 ± 0.26 [e] | 8.27 ± 0.33 [d] | 6.23 ± 0.14 [a] | 8.32 ± 0.18 [d] | 7.34 ± 0.53 [b] | 7.81 ± 0.77 [c] |
| General taste | 8.62 ± 0.15 [f] | 8.10 ± 0.21 [d] | 5.82 ± 0.15 [a] | 8.36 ± 0.22 [e] | 7.25 ± 0.84 [b] | 7.92 ± 0.30 [c] |
| Bitterness | 8.01 ± 0.24 [c] | 8.21 ± 0.28 [c] | 6.52 ± 0.35 [a] | 8.43 ± 0.11 [d] | 8.21 ± 0.36 [c] | 7.41 ± 0.24 [b] |
| Carbonation | 8.81 ± 0.11 [d] | 8.40 ± 0.18 [bc] | 8.31 ± 0.11 [a] | 8.54 ± 0.09 [c] | 8.52 ± 0.18 [bc] | 8.40 ± 0.19 [ab] |
| Body | 8.52 ± 0.15 [e] | 8.02 ± 0.25 [c] | 6.06 ± 0.95 [a] | 8.21 ± 0.24 [d] | 7.31 ± 0.34 [b] | 7.22 ± 0.95 [b] |
| Mouthfeel | 8.33 ± 0.27 [e] | 7.91 ± 0.53 [c] | 6.22 ± 0.77 [a] | 8.08 ± 0.57 [c] | 7.13 ± 0.25 [b] | 7.01 ± 0.64 [b] |
| General acceptability | 8.42 ± 0.21 [e] | 8.01 ± 0.20 [d] | 6.41 ± 0.82 [a] | 8.12 ± 0.36 [d] | 7.12 ± 0.33 [b] | 7.92 ± 0.77 [c] |

Data are expressed as mean ± standard deviation. [a–f]—mean values in the same column followed by a different letter are statistically different ($p < 0.05$).

### 3.3. Principal Component Analysis of the Beer Samples and Their Characteristics

Principal component analysis (PCA) loadings of the physicochemical and sensory characteristics and beer samples are shown in Figure 5. The two axes PC1 and PC2 show 44.84% and 30.22% total variance. The first plot, PC1, shows highly significant correlations ($p < 0.05$) between physicochemical characteristics of real extract and alcohol content (r = 0.821), between density and apparent extract (r = 0.997), bitterness and pH (r = 0.873),

and between physicochemical and sensory characteristics carbon dioxide and carbonation (r = 0.985). During alcohol fermentation, the fermentable sugars were transformed into alcohol and therefore the real extract was well correlated with the beer alcohol content. The beer sensory carbonation gave the panelists a burning sensation on the tongue, determined by the amount of carbon dioxide [78]. All the sensory characteristics of body, general acceptability, aroma, general taste, bitterness, and color were grouped together on the bottom right of the PCA graph. This may be explainable, since on general acceptability of the beers, the highest impact was on the beer's taste [79]. The taste was significant correlated ($p < 0.05$) with bitterness (r = 0.888), aroma (r = 0.995), body (r = 0.954), mouthfeel (r = 0.925), and general acceptability (r = 0.988). Unexpectedly, PC2 clearly distinguished between the physical and sensory characteristics color and bitterness. This was probably due to the raw materials used in this study. The chemical bitterness value was mostly affected by hop addition, whereas the sensory one was affected by beer composition. The sorghum had a sweet taste, which may have affected the bitterness perception of the panelists. Additionally, the sensory color value evaluation may have affected panelists' perception of the traditional blonde beer color from barley malt and therefore it may be not be positively correlated with the physical characteristic. Of all the analyzed samples, the CS seemed to be rated closest to the S3 sample, which are situated at the right top of the PCA graph. These data are in agreement with those reported by Schnitzenbaumer et al. [9], who also concluded that beer with 40% unmalted sorghum is the closest to beer with 100% barley malt. Both S3 and CS beer samples were closely rated on the physicochemical characteristics of alcohol and carbon dioxide, and sensory ones of carbonation, mouthfeel, and appearance. These characteristics seem to have had the highest impact on beer consumers' preferences. Mouthfeel is closely related with the sensory characteristics of carbonation, drying, astringency, fullness, wateriness, warming, smoothness, and mouth-coating. Different beer compounds have been reported to affect the mouthfeel of beer, which include the content of ethanol, glycerol, polyphenol, carbon dioxide, and foam structure [80]. Therefore, the association on the PCA graph between physicochemical characteristics of alcohol and carbon dioxide and the sensory one of mouthfeel was somehow predictable. As may be seen, the beers most preferred by the panelists (CS, S3, S1) are situated along PC2 axes on the right side of the PCA graph. All the sensory characteristics were closely associated by them. Among the physicochemical characteristics, the ones with the highest impact were the alcohol content, real extract, and carbonation values. The real extract is a very important factor in beer sensory characteristics. Its composition influences the beer's body, carbonation, and flavor. Beer is a drink that is distinguished from other carbonated drinks by the ability to form foam with some persistence. Therefore, the carbonation is a very important sensory characteristic. High carbon dioxide content in the beer gives foam, which is constantly fed from the bottom and dries on the surface, where it becomes stable. The consequence is the increase in the persistence of the foam. It can be promoted by stirring the beer, which facilitates the access of air, forming more fine and stable bubbles in the beer. Therefore, the physicochemical carbon dioxide characteristic had a high impact on panelists' beer preference. Beers S4 and S5 are also clustered together on the bottom left side on the PCA graph, indicating that the samples with high amounts of unmalted sorghum in their recipe were similar from the physicochemical and sensory points of view. These samples were most associated with turbidity and oxygen values. This association may be due to the fact that these samples contained the lowest amounts of malt in their recipe. S1, which was also well appreciated by the panelists, who rated it as "like very much" according to the mean value of general acceptability, which was closely associated with sensory characteristics of color, bitterness, general taste, aroma, body, and general acceptability. The plot of PC1 vs. PC2 clearly distinguished the S2 sample by all sensory characteristics, in agreement with our sensory data, which indicates that this sample was the least appreciated by the panelists.

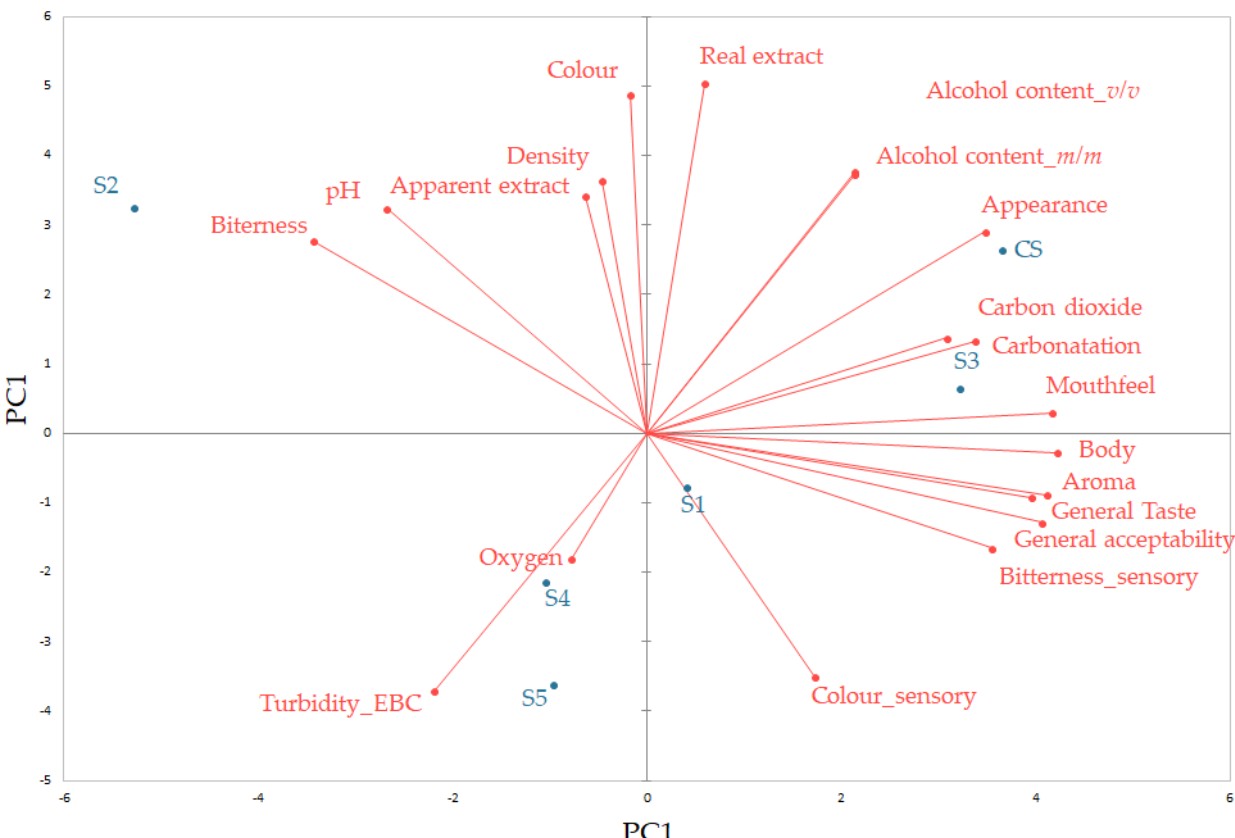

**Figure 5.** Principal component analysis of the beer samples and their physicochemical and sensory characteristics.

## 4. Conclusions

Many countries throughout the world have demonstrated their ability to produce different sorghum beers and malts. The manufacturing of beer from this cereal is a practical alternative in the present, when climate change is becoming increasingly noticeable in our country, while also being a novelty on the profile market in Romania.

Sorghum, commonly known as the camel plant, is one of the most adaptable food crops and will be crucial for a sustainable economy, since it is more drought tolerant than other cereal crops. The potential of this grain to be used as a raw material in the beer industry was demonstrated through laboratory research, and a variant of the finished product was also obtained that met the qualification of very good, meaning it did not present any types of deficiencies or obvious defects. In terms of the key physicochemical characteristics, the beer made with 60% sorghum malt and 40% unmalted sorghum with 4% Termamyl Classic enzyme performed the best out of all the recipes studied: its original extract was 11.26% $m/m$, apparent extract was 4.59% $m/m$, alcohol content was 4.12% ($v/v$), and turbidity was 0.74 EBC. The sample that tasted the best was made with 60% sorghum malt and 40% unmalted sorghum with 4% Termamyl Classic enzyme. Our goal was to develop a final beer with good physicochemical characteristics, consumer appeal, and inexpensive production costs. We succeeded in achieving these objectives. Future research is intended to improve manufacturing methods and technological processes to enhance the sensory qualities of the finished beer. For the effective implementation of pertinent solutions with an impact on the final product's quality, this process requires focused and ongoing efforts in research and coordination of all involved parties. These results demonstrate that industry and academic research can foster innovation in the production of new beer varieties using sorghum in an organized manner, assisting in improving product quality and lowering

quality deviations. The technical, economic, and customer acceptability of the new product assortments will all be taken into account in technological innovation.

**Author Contributions:** Conceptualization, M.E.C., A.D. and G.G.C.; methodology, A.C. and R.V.S.; formal analysis, Á.A., A.C. and A.D.; investigation, M.E.C., R.V.S. and Á.A.; resources, G.G.C.; writing—original draft preparation, A.D., R.V.S. and M.E.C.; writing—review and editing, A.C. and A.D. All authors have read and agreed to the published version of the manuscript.

**Funding:** The work of the author Marius Eduard Ciocan was supported by the project "PROIN-VENT", POCU/993/6/13-Code 153299, financed by The Human Capital Operational Programme 2014–2020 (POCU), Romania. This work was funded by the Ministry of Research, Innovation and Digitalization within Program 1—Development of national research and development system, Subprogram 1.2—Institutional Performance—RDI excellence funding projects, under contract no. 10PFE/2021.

**Institutional Review Board Statement:** Not applicable.

**Informed Consent Statement:** Not applicable.

**Data Availability Statement:** The data presented in this study are available in this article.

**Conflicts of Interest:** The authors declare no conflict of interest.

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
