# Peer review of "Brewing with Unmalted and Malted Sorghum: Influence on Beer Quality"

_fermentation, doi:10.3390/fermentation9050490_

Round 1

Reviewer 1 Report

Titleà Brewing with Unmalted and Malted Sorghum: Influence on Beer Quality

Line 23à quality results in which terms? Brewing characteristics or sensory attributes?

Line 37à Why raw materials on the local market leads to the improvement of the content in bioactive compounds? Please add a reference or explain in more detail.

Line 84à Why very short shelf life of 24-72 hours? Due to an higher pH value? Less alcohol content? Please give an explanation.

Line 214à sensory attributes were evaluated in this order? In general, it is preferred to evaluate first overall liking, then appearance, odour, taste, flavour and mouthfeel.

Line 215à information regarding the washing procedure are already reported at line 211

Line 264à pH of the wort?

Table 4à please, in the table caption, indicate the composition of each sample, report in full the composition of each sample ID

Line 332à 95°C and 60°C??

Line 333à in the odour, taste

Figure 3à please, modify the name of the independent variable B, change sorghum flour in unmalted sorghum in order to be consistent with M&M section and the title of the article

Line 409à please change unmalted sorghum and not sorghum flour

Author Response

Dear Referee,  

We would like to thank the referee for the close reading and for the proper suggestions.

We hope that we provide all the answers to the reviewer’s comments.

Thank you very much for the recommendations to publish our paper entitled “Brewing with Unmalted and Malted Sorghum: Influence on Beer Quality”.

The present version of the paper has been revised according to the reviewer’s suggestions.       

We uploaded the corrected version of the article for which we used the red colour for the addition text.

Kindly take notice of these changes made to the manuscript's text. The red colour indicates the answer to your suggestions, while the blue colour indicates the responses to the recommendations of the other reviewer, and the green colour indicates the corrections made to the editor's recommendations. We regret not include the responses to your suggestions in this letter. The paper (V2) has been attached with all the revisions made per the suggestions of the reviewers.

Point 1: Line 23 - quality results in which terms? Brewing characteristics or sensory attributes?

Response 1: First of all, we would like to thank the referee for the close reading and for all the given comments suitable for improving the manuscript.

We made the changes according to the referee suggestions. 

Point 2: Line 37 -  Why raw materials on the local market leads to the improvement of the content in bioactive compounds? Please add a reference or explain in more detail.

Response 2: We would like to thank to the referee for her/his remarks. We made the changes according to the referee suggestions.

Point 3: Line 84 - Why very short shelf life of 24-72 hours? Due to an higher pH value? Less alcohol content? Please give an explanation.

Response 3: We would like to thank to the referee for her/his remarks. We have explained in the text that it is about craft beer that is not filtered or stabilized. This beer is consumed in a state of active fermentation, so it can have a higher acidity and much lower alcohol content.  

Point 4: Line 214 - sensory attributes were evaluated in this order? In general, it is preferred to evaluate first overall liking, then appearance, odour, taste, flavour and mouthfeel.

Response 4: We would like to thank to the referee for her/his remarks. We made the changes according to the referee suggestions.

Point 5: Line 215 - information regarding the washing procedure are already reported at line 211.

Response 5: We would like to thank to the referee for her/his remarks. We corrected it by excluding this text.

Point 6: Line 264 - pH of the wort?

Response 6: Response 8: We would like to thank to the referee for her/his remarks. We corrected it.

Point 7: Table 4 - please, in the table caption, indicate the composition of each sample, report in full the composition of each sample ID

Response 7: We would like to thank to the referee for her/his remarks. We indicate the composition of each sample ID.

Point 8: Line 332 -  95°C and 60°C??

Response 8: We would like to thank to the referee for her/his remarks. We corrected it.

Point 9: Line 333 - in the odour, taste

Response 9: We would like to thank to the referee for her/his remarks. We made the changes according to the referee suggestions. 

Point 10: Figure 3 - please, modify the name of the independent variable B, change sorghum flour in unmalted sorghum in order to be consistent with M&M section and the title of the article.

Response 10: We would like to thank to the referee for her/his remarks. We made the changes according to the referee suggestions.

 Point 11: Line 409 - please change unmalted sorghum and not sorghum flour

Response 11: We would like to thank to the referee for her/his remarks. We made the changes according to the referee suggestions.

Reviewer 2 Report

The manuscript studied different portions of malted and unmalted sourghum for beer production. Physicochemical and sensorial parameters of the obtained beers were evaluated. The manuscript is well described and the subject is relevant. However, some issue may be clarified in order to improve the quality of the study.

In the abstract, the authores conclude that the formulation containing 60% sorghum malt and 40% unmalted sorghum showed the high quality results. However, no data was included in the abstract. Please, add more details about what the authors considered high quality results. Furthermore, they conclude that "The resulting sorghum beer typically has low alcohol content, a complex, aromatic, slightly sour flavor, a mild bitter or astringent sensation, less stable foam, and a short shelf life.", but not all of these parameters were evaluated. Please, revise .

Material and methods, section 2.3.3 Sensory analysis. Please, describe what semi-trained evaluators mean. Add more information about the panelists, eg. age, sex, how they were recruted, if they are regular consumers of beers.

In the Results and Discussion section, add statistical analysis of data presented in Tables 4 and 5. Further, revise statistical analysis of data from Table 7. It is interesting to compare the data among the different formulations.

The PCA analysis showed interesting clustering of the samples, and the authors can enhance the discussion based on the physicochemical analysis values ​​that were related to the preferred samples

In the conclusion section, lines 436 and 437 the authors described "The sample that tasted the best was made entirely of unmalted buckwheat and contained 2% Termamyl Classic enzyme." I did not found this formulation in the work. Please, revise.

Author Response

Response to Reviewer 2

Dear Referee,  

We would like to thank the referee for the close reading and for the proper suggestions.

We hope that we provide all the answers to the reviewer’s comments.

Thank you very much for the recommendations to publish our paper entitled “Brewing with Unmalted and Malted Sorghum: Influence on Beer Quality”.

The present version of the paper has been revised according to the reviewer’s suggestions.       

We uploaded the corrected version of the article for which we used the blue colour for the addition text.

Kindly take notice of these changes made to the manuscript's text. The blue colour indicates the answer to your suggestions, while the red colour indicate the responses to the recommendations of the other reviewer, and the green colour indicates the corrections made to the editor's recommendations. We regret not include the responses to your suggestions in this letter. The paper (V2) has been attached with all the revisions made per the suggestions of the reviewers.

Point 1: In the abstract, the authors conclude that the formulation containing 60% sorghum malt and 40% unmalted sorghum showed the high quality results. However, no data was included in the abstract. Please, add more details about what the authors considered high quality results. Furthermore, they conclude that "The resulting sorghum beer typically has low alcohol content, a complex, aromatic, slightly sour flavor, a mild bitter or astringent sensation, less stable foam, and a short shelf life.", but not all of these parameters were evaluated. Please, revise .

Response 1: First of all, we would like to thank the referee for the close reading and for all the given comments suitable for improving the manuscript.

We made the changes according to the referee suggestions.

Point 2: Material and methods, section 2.3.3 Sensory analysis. Please, describe what semi-trained evaluators mean. Add more information about the panelists, eg. age, sex, how they were recruted, if they are regular consumers of beers.

Response 2: We would like to thank to the referee for her/his remarks. We made the changes according to the referee suggestions.

Point 3: In the Results and Discussion section, add statistical analysis of data presented in Tables 4 and 5. Further, revise statistical analysis of data from Table 7. It is interesting to compare the data among the different formulations.

Response 3: We would like to thank to the referee for her/his remarks. We made the changes according to the referee suggestions.

Point 4: The PCA analysis showed interesting clustering of the samples, and the authors can enhance the discussion based on the physicochemical analysis values ​​that were related to the preferred samples.

Response 4: We would like to thank to the referee for her/his remarks. We made the changes according to the referee suggestions. We added the discusion based on the physicochemical analysis values ​​that were related to the preferred samples.

 Point 5: In the conclusion section, lines 436 and 437 the authors described "The sample that tasted the best was made entirely of unmalted buckwheat and contained 2% Termamyl Classic enzyme." I did not found this formulation in the work. Please, revise.

Response 5: We would like to thank to the referee for her/his remarks. We made the changes according to the referee suggestions. We revised it.

Round 2

Reviewer 1 Report

Thank you for providing proper corrections to the text.
I have no other comment.